# Post COVID-19 condition after Wildtype, Delta, and Omicron SARS-CoV-2 infection and prior vaccination: Pooled analysis of two population-based cohorts

Tala Ballouz[1], Dominik Menges[1], Marco Kaufmann[1], Rebecca Amati[2], Anja Frei[1], Viktor von Wyl[1], Jan S. Fehr[1], Emiliano Albanese[2], Milo A. Puhan[1]*

1 Epidemiology, Biostatistics and Prevention Institute (EBPI), University of Zurich (UZH), Zurich, Switzerland,
2 Faculty of Biomedical Sciences, Institute of Public Health (IPH), Università della Svizzera Italiana (USI), Lugano, Switzerland

☯ These authors contributed equally to this work.
* miloalan.puhan@uzh.ch

**Data Availability Statement:** All relevant data are within the paper and its Supporting Information files.

## Abstract

### Background

Post COVID-19 condition (PCC) is an important complication of SARS-CoV-2 infection, affecting millions worldwide. This study aimed to evaluate the prevalence and severity of post COVID-19 condition (PCC) with novel SARS-CoV-2 variants and after prior vaccination.

### Methods

We used pooled data from 1350 SARS-CoV-2-infected individuals from two representative population-based cohorts in Switzerland, diagnosed between Aug 5, 2020, and Feb 25, 2022. We descriptively analysed the prevalence and severity of PCC, defined as the presence and frequency of PCC-related symptoms six months after infection, among vaccinated and non-vaccinated individuals infected with Wildtype, Delta, and Omicron SARS-CoV-2. We used multivariable logistic regression models to assess the association and estimate the risk reduction of PCC after infection with newer variants and prior vaccination. We further assessed associations with the severity of PCC using multinomial logistic regression. To identify groups of individuals with similar symptom patterns and evaluate differences in the presentation of PCC across variants, we performed exploratory hierarchical cluster analyses.

### Results

We found strong evidence that vaccinated individuals infected with Omicron had reduced odds of developing PCC compared to non-vaccinated Wildtype-infected individuals (odds ratio 0.42, 95% confidence interval 0.24–0.68). The odds among non-vaccinated individuals were similar after infection with Delta or Omicron compared to Wildtype SARS-CoV-2. We

**Funding:** This study is part of the Corona Immunitas research network, coordinated by the Swiss School of Public Health (SSPH+), and funded by fundraising of SSPH+ including funds of the Swiss Federal Office of Public Health and private funders (ethical guidelines for funding stated by SSPH+ were respected), by funds of the cantons of Switzerland (Vaud, Zurich, and Basel), and by institutional funds of the Universities. Additional funding specific for the two cohorts included in this study was received from the Department of Health of the canton of Zurich, the University of Zurich (UZH) Foundation, and the Swiss Federal Office of Public Health. TB received funding from the European Union's Horizon 2020 research and innovation programme under the Marie Skłodowska-Curie grant agreement No 801076, through the SSPH+ Global PhD Fellowship Programme in Public Health Sciences (GlobalP3HS) of the SSPH+. DM received funding by the UZH Postdoc Grant, grant no. FK-22-053. The funders had no role in study design, data collection and analysis, decision to publish, or preparation of the manuscript.

**Competing interests:** The authors have declared that no competing interests exist.

found no differences in PCC prevalence with respect to the number of received vaccine doses or timing of last vaccination. The prevalence of PCC-related symptoms among vaccinated, Omicron-infected individuals was lower across severity levels. In cluster analyses, we identified four clusters of diverse systemic, neurocognitive, cardiorespiratory, and musculoskeletal symptoms, with similar patterns across variants.

## Conclusion

The risk of PCC appears to be lowered with infection by the Omicron variant and after prior vaccination. This evidence is crucial to guide future public health measures and vaccination strategies.

## Introduction

Post COVID-19 condition (PCC) is an important complication of SARS-CoV-2 infection, posing a substantial burden on health care systems worldwide [1,2]. Population-based studies have estimated that about 20–30% of non-vaccinated individuals with a confirmed Wildtype SARS-CoV-2 infection develop PCC [3], defined as symptoms persisting beyond three months after infection and not explained by an alternative diagnosis [1]. As of writing, half a billion SARS-CoV-2 infections have been diagnosed worldwide [4], and more than 144 million people are estimated to have been affected by PCC in 2020 and 2021 [2]. With the global implementation of vaccination campaigns and the emergence of novel SARS-CoV-2 variants of concern, future pandemic management will depend on the incidence of PCC in vaccinated individuals infected with novel variants.

Current evidence suggests that the risk of PCC is relevantly reduced by vaccination with SARS-CoV-2 vaccination [5]. Several studies based on different designs and populations found that the risk of PCC is approximately halved in vaccinated compared to non-vaccinated individuals [6–16]. While most evaluated symptoms persisting for more than four [7,8,16] to 12 weeks [9,11,12,14,15] after SARS-CoV-2 infection, few assessed symptoms beyond six months and found a lower [6] or no [10,13] risk reduction with vaccination.

Meanwhile, the evidence regarding infection with novel SARS-CoV-2 variants remains limited and inconsistent. Some evidence suggests that infections with the Omicron variant results in lower risk of developing PCC compared to the Delta [17] or any previous variant [18]. One study demonstrated a reduced risk with the Alpha variant, but not with Delta or Omicron compared to Wildtype SARS-CoV-2 [8]. Another study reported a higher risk of PCC and greater PCC-related symptom burden with Wildtype SARS-CoV-2 compared to the Alpha and Delta variants [19]. However, all four studies included specific samples not representative of the general population, and only one evaluated PCC beyond 12 weeks after infection [19]. Therefore, more knowledge regarding the expected risk reduction through infection with newer variants and preventive effects of vaccination on PCC in the longer-term is urgently needed.

This study aimed to evaluate the prevalence and severity of PCC in individuals infected by Wildtype, Delta, and Omicron SARS-CoV-2 with and without prior vaccination. Specific objectives were to assess the difference in risk of developing PCC with emerging variants and vaccination, evaluate changes in symptom severity, and identify prevalent symptom clusters and their evolution across pandemic waves.

## Methods

### Study design and participants

This study is based on pooled data from two population-based cohorts in Switzerland (S1 Table). The Zurich SARS-CoV-2 Cohort is a prospective longitudinal cohort of 1106 SARS-CoV-2-infected individuals, recruited shortly after infection based on an age-stratified random sample of all diagnosed cases between Aug 6, 2020, and Jan 19, 2021, through the Department of Health of the canton of Zurich, Switzerland [20,21]. The study was pre-registered (https://doi.org/10.1186/ISRCTN14990068) and approved by the ethics committee of the canton of Zurich (BASEC 2020–01739). The Corona Immunitas seroprevalence study is a prospective longitudinal cohort of an age-stratified random population sample derived through the Swiss Federal Statistical Office [22,23]. For this study, we leveraged data from the fifth phase of Corona Immunitas including 1844 participants from the cantons of Zurich and Ticino, Switzerland, for which baseline assessments took place between Mar 1–31, 2022 [24]. The study was registered (https://doi.org/10.1186/ISRCTN18181860) and approved by the ethics committees of the cantons of Zurich (BASEC 2020–01247) and Ticino (BASEC 2020–01514). All participants provided electronic (Zurich SARS-CoV-2 Cohort) or written (Corona Immunitas) informed consent prior to participation.

### Data collection and pooling

We collected data using electronic questionnaires, managed through the Research Electronic Data Capture (REDCap) platform [25,26]. In both cohorts, participants completed baseline and regular follow-up questionnaires (S1 Table) [20–24]. Questionnaires were closely aligned and used the same wording for critical questions related to sociodemographic characteristics, current symptoms, and current health status. The presence of PCC-related symptoms was elicited using a list of 23 symptoms frequently reported in the literature (S2 Table). From the Zurich SARS-CoV-2 Cohort, we included all participants who had completed follow-up at six months after SARS-CoV-2 infection (a range of five to seven months after infection was allowed). From Corona Immunitas, we included all participants who, either at baseline or during follow-up, reported having last been infected by SARS-CoV-2 during the timeframes of predominance of the Delta and Omicron variants, and completed the baseline or either of the follow-up assessments approximately six (five to seven) months after the most recent reported infection (S1 Fig).

### Outcome definition

We defined the primary outcome of PCC as symptoms present within the last seven days at six months after the most recent diagnosed SARS-CoV-2 infection, reported by participants to be related to COVID-19. This outcome definition has previously been shown to likely adequately capture the presence of PCC at a population level [21]. Secondary outcomes were individual PCC-related symptoms and severity of PCC at six months, assessed based on symptom count at follow-up (stratified into groups with 1–2, 3–5, and ≥6 symptoms). In sensitivity analyses, we categorised severity based on the EuroQoL visual analogue scale (EQ-VAS) and previously applied cut-offs [21,27,28] as having mild (EQ-VAS >70), moderate (EQ-VAS 51–70), or severe (EQ-VAS ≤50) PCC.

In lack of viral samples for genetic analysis of SARS-CoV-2 variants, we determined the most likely infecting variant based on reported infection dates. We categorised all Zurich SARS-CoV-2 Cohort participants as infected by Wildtype SARS-CoV-2. In accordance with viral predominance (≥80% of diagnosed infections) in Switzerland, we determined infections

between Jul 7, 2021, and Dec 31, 2021, as most likely due to Delta, and infections from Jan 1, 2022, as most likely due to Omicron SARS-CoV-2 [29]. We used the date of the first positive SARS-CoV-2 test as infection date, while positive tests more than 60 days before were considered a separate prior infection.

## Statistical analyses

We descriptively analysed population characteristics and the prevalence of PCC, overall and stratified by SARS-CoV-2 variants, prior vaccination, and symptom severity. We calculated 95% Wilson confidence intervals (CIs) for proportions. We used multivariable logistic regression models to evaluate associations of infection during different variant timeframes and prior vaccination with PCC. In the primary analysis model, we combined variant and vaccination status information to evaluate the joint association within vaccinated or non-vaccinated, Delta- or Omicron-infected individuals compared to non-vaccinated, Wildtype-infected individuals, while adjusting for prior infection. Model selection was based on prior knowledge (a priori variables: age, sex, presence of comorbidities (any of hypertension, diabetes, cardiovascular disease, chronic respiratory disease, malignancy, or immune suppression), and initial hospitalization due to COVID-19) and the Akaike and Bayesian Information Criteria (no further variables added). In sensitivity analyses and to ensure comparability with other studies [8,18], we used an alternative model including SARS-CoV-2 variant, vaccination status, and prior infection as separate variables. We additionally estimated the absolute risk reduction (i.e., adjusted risk differences) based on a logistic regression model using an identity link [30]. Furthermore, we conducted multivariable multinomial logistic regression analyses to evaluate the association of different strains with the severity of PCC, including sensitivity analyses using alternative definitions for PCC severity.

In exploratory analyses, we assessed the presence of specific clusters of PCC-related symptoms at six months using multiple correspondence combined with hierarchical cluster analyses [31]. Sixteen long-term symptoms reported by at least five percent of participants with PCC were included, on which a multiple correspondence analysis was performed. We retained dimensions that explained 90% of the variance and performed an agglomerative hierarchal clustering on the selected dimensions using Ward minimum-variance linkage methods [31,32]. We based the selection of the number of clusters on findings from other studies [33–35] and by maximizing the relative loss of inertia. We selected four clusters for the main analysis, but present findings from sensitivity analyses assuming five and six clusters for future comparison. We descriptively evaluated PCC-related symptoms and participant characteristics across clusters to identify factors potentially associated with membership in each of the clusters.

We performed all analyses in R (version 4.0.3).

## Results

### Participant characteristics

We included data from 1045 Zurich SARS-CoV-2 Cohort participants and 305 Corona Immunitas participants reporting a SARS-CoV-2 infection with six months of follow-up (Table 1 and S1 Fig and S3 and S4 Tables). Median follow-up was 183 days (interquartile range [IQR] 182–186 days) across all participants. Zurich SARS-CoV-2 Cohort participants were slightly older on average, with a median age of 51 years (IQR 35–66) compared to 43 years (30–54) among Corona Immunitas participants. The proportion of female participants was 50.7% (n = 530) and 58.7% (n = 179), respectively. Zurich SARS-CoV-2 Cohort participants more frequently reported the presence of at least one medical comorbidity (29.5%) compared to

**Table 1. Participant characteristics of Wildtype, Delta, or Omicron SARS-CoV-2-infected individuals from the Zurich SARS-CoV-2 Cohort and the Corona Immunitas Phase 5 seroprevalence studies.**

| | Zurich SARS-CoV-2 Cohort | Corona Immunitas | Overall |
|---|---|---|---|
| | (N = 1045) | (N = 305) | (N = 1350) |
| **Timeframe of diagnosis** | Aug 5, 2020 –Jan 19, 2021 | Jul 15, 2021 –Feb 25, 2022 | Aug 5, 2020 –Feb 25, 2022 |
| **Median follow-up (IQR; days)** | 183.5 (182–186) | 181 (164–196) | 183 (182–186) |
| **Age, median (IQR)** | 51 (35–66) | 43 (30–54) | 48 (34–63) |
| **Female sex** | 530 (50.7%) | 179 (58.7%) | 709 (52.5%) |
| **Presence of chronic comorbidity** | 308 (29.5%) | 43 (14.1%) | 351 (26.0%) |
| **Smoking status** | | | |
| Non-smoker | 625 (60.1%) | 201 (65.9%) | 826 (61.4%) |
| Ex-smoker | 282 (27.1%) | 62 (20.3%) | 344 (25.6%) |
| Smoker | 133 (12.8%) | 42 (13.8%) | 175 (13.0%) |
| *Missing* | *5* | *0* | *5* |
| **Body mass index, median (IQR; kg/m2)** | 24.2 (21.9–26.6) | 23.3 (21.5–26.0) | 24.0 (21.7–26.5) |
| **Highest education** | | | |
| None or mandatory school | 41 (3.9%) | 18 (5.9%) | 59 (4.4%) |
| Vocational training or specialised baccalaureate | 438 (42.2%) | 154 (50.8%) | 592 (44.1%) |
| Higher technical school or college | 276 (26.6%) | 36 (11.9%) | 312 (23.2%) |
| University | 284 (27.3%) | 95 (31.4%) | 379 (28.2%) |
| *Missing* | *6* | *2* | *8* |
| **Employment status** | | | |
| Employed | 668 (64.2%) | 57 (18.8%) | 725 (53.9%) |
| Retired | 256 (24.6%) | 181 (59.5%) | 437 (32.5%) |
| Student | 50 (4.8%) | 32 (10.5%) | 82 (6.1%) |
| Unemployed or other | 67 (6.4%) | 34 (11.2%) | 101 (7.5%) |
| *Missing* | *4* | *1* | *5* |
| **Hospitalised due to COVID-19** | 44 (4.2%) | 2 (0.7%) | 46 (3.4%) |
| **SARS-CoV-2 variant** | | | |
| Wildtype | 1045 (100%) | 0 (0.0%) | 1045 (77.4%) |
| Delta | 0 (0.0%) | 99 (32.5%) | 99 (7.3%) |
| Omicron | 0 (0.0%) | 206 (67.5%) | 206 (15.3%) |
| **Prior vaccination** | 0 (0.0%) | 232 (77.1%) | 232 (17.2%) |
| **Vaccine doses[a]** | | | |
| 1–2 doses | – | 173 (74.6%) | 173 (74.6%) |
| 3 doses | – | 59 (25.4%) | 59 (25.4%) |
| **Time since last vaccine dose[a]** | | | |
| <6 months | – | 180 (77.6%) | 180 (77.6%) |
| ≥6 months | – | 52 (22.4%) | 52 (22.4%) |
| **Type of vaccines received[a]** | | | |
| mRNA | – | 231 (99.6%) | 231 (99.6%) |
| Adenovirus vector | – | 1 (0.4%) | 1 (0.4%) |
| **Prior SARS-CoV-2 infection** | 0 (0.0%) | 36 (11.8%) | 36 (2.7%) |

**Legend**: BMI = body mass index, IQR = interquartile range.

[a] Percentages among those that have received at least one vaccine dose prior to infection.

Corona Immunitas (14.1%). 232 (77.1%) Corona Immunitas participants reported to have received at least one COVID-19 vaccine prior to infection, among which 173 (74.6%) had received one or two doses, and 59 (25.4%) had received three doses. Almost all (n = 231,

99.6%) had received a mRNA-based vaccine (i.e., BNT162b2 or mRNA-1273), while one participant had received a vector-based vaccine (i.e., JNJ-78436735). A previous SARS-CoV-2 infection was reported by 36 (11.8%) Corona Immunitas participants.

## Association of variants and vaccination with post COVID-19 condition

Overall, 25.3% (95% CI 22.7–28.0%, n = 264) of individuals infected with Wildtype SARS-CoV-2, 17.2% (11.0–25.8%, n = 17) of Delta-infected, and 13.1% (9.2–18.4%, n = 27) of Omicron-infected individuals had PCC six months after infection. The proportion of participants with PCC among non-vaccinated individuals infected with the Delta (21.6%, 11.4–37.2%, n = 8) and Omicron (21.9%, 11.0–38.8%, n = 7) variants were similar to Wildtype infection without prior vaccination. Among vaccinated individuals, 14.8% (8.0–25.7%, n = 9) had PCC after Delta, and 11.1% (7.2–16.7%, n = 19) after Omicron infection. We observed no clear patterns in PCC-related symptoms across individuals infected with different variants (S2 Fig).

When assessing the association between infection with different SARS-CoV-2 variants and prior vaccination with PCC at six months, there was strong evidence for a reduction in the odds among vaccinated individuals infected by the Omicron variant (odds ratio [OR] 0.42, 95% CI 0.24–0.68, p = 0.0008) compared to non-vaccinated, Wildtype SARS-CoV-2-infected individuals, based on multivariable logistic regression analyses adjusted for age, sex, presence of comorbidities, hospitalization due to COVID-19, and prior infection (Fig 1). The estimated absolute risk reduction for PCC was -10.6% (-16.2% to -5.0%). Meanwhile, there was insufficient evidence for a reduction in the odds among vaccinated, Delta-infected individuals (OR 0.55, 0.25–1.08, p = 0.11), and among non-vaccinated individuals infected by the Delta (OR 0.84, 0.34–1.87, p = 0.69) or Omicron (OR 0.87, 0.33–2.06, p = 0.77) variant. The

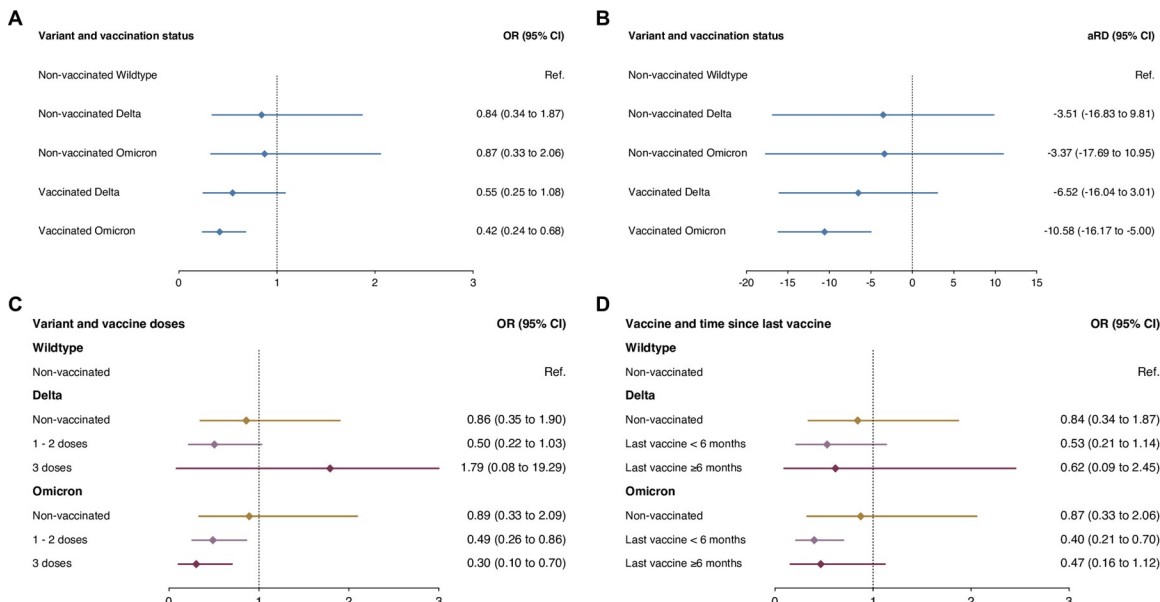

**Fig 1. Association of Delta and Omicron SARS-CoV-2 infection and prior vaccination with post COVID-19 syndrome six months after SARS-CoV-2 infection.** Panels **A** and **B** show odds ratios and adjusted risk differences among non-vaccinated and vaccinated individuals infected with the Delta or Omicron variant compared to non-vaccinated individuals infected by SARS-CoV-2, based on multivariable logistic regression models adjusted for age, sex, presence of comorbidities, initial hospitalization due to COVID-19, and prior infection. Panels **C** and **D** show odds ratios for having received one or two vaccine doses or three doses prior to infection and for having been vaccinated less than six months prior or six or more months prior to infection with the Delta or Omicron variants, based on multivariable logistic regression models adjusted for age, sex, presence of comorbidities, initial hospitalization due to COVID-19, and prior infection. Legend: aRD = adjusted risk difference, CI = confidence interval, OR = odds ratio, Ref. = reference group.

corresponding absolute risk reduction was -6.5% (-16.0% to 3.0%) for vaccinated individuals infected with Delta, -3.5% (-16.8% to 9.8%) for non-vaccinated, Delta-infected individuals, and -3.4% (-17.7% to 11.0%) for non-vaccinated, Omicron-infected individuals. In sensitivity analyses including variant and vaccination as separate variables, results were similar for associations of Delta (OR 0.92, 0.44–1.83, p = 0.83) and Omicron (OR 0.78, 0.35–1.65, p = 0.54) infection, as well as prior vaccination with PCC (OR 0.55, 95% CI 0.26–1.19, p = 0.12; S3 Fig).

In analyses stratified by the number of vaccine doses received prior to infection, there was no evidence for a difference in the odds of PCC between individuals having received 1–2 vaccine doses and individuals having received three doses among individuals infected with the Delta variant, while there was a tendency for a stronger association with three (OR 0.30, 0.10–0.70, p = 0.012) compared to 1–2 doses (OR 0.49, 0.26–0.86, p = 0.019) among Omicron-infected individuals (Fig 1). The odds of PCC independent of variant was equally reduced for both 1–2 doses and three doses compared to non-vaccinated individuals, however with high uncertainty (S3 Fig). When evaluating the timing since the last vaccine dose at infection, there was no evidence for a difference between individuals that were vaccinated six or more months prior, and individuals vaccinated less than six months prior to infection both after Delta and Omicron infection (Fig 1). Findings were similar in a sensitivity analysis of associations independent of variant (S3 Fig).

## Symptom severity

Regarding the severity of PCC-related symptoms at six months, there were similar patterns across groups infected by different variants and with and without prior vaccination (Fig 2). In accordance with the overall prevalence of PCC, we observed lower prevalences of individuals reporting 1–2, 3–5, and ≥6 PCC-related symptoms among vaccinated, Omicron-infected individuals compared to other groups. Sensitivity analyses using different severity definitions resulted in similar findings (S4 Fig). While symptom prevalence among vaccinated, Omicron-infected individuals was lower for those reporting ≥3 symptoms, 4% (n = 7) still reported such symptoms after six months. In multivariable multinomial logistic regression analyses, there was strong evidence for a reduction in the odds among those reporting 1–2 symptoms among vaccinated, Omicron-infected individuals compared to non-vaccinated, Wildtype-infected individuals (OR 0.39, 95% CI 0.21–0.74, p = 0.0038), while the statistical evidence for a reduction in the odds among those reporting 3–5 (OR 0.44, 0.16–1.19, p = 0.11) and ≥6 symptoms (OR 0.50, 95% CI 0.11–2.21, p = 0.36) was insufficient (Tables 2 and S5–S7).

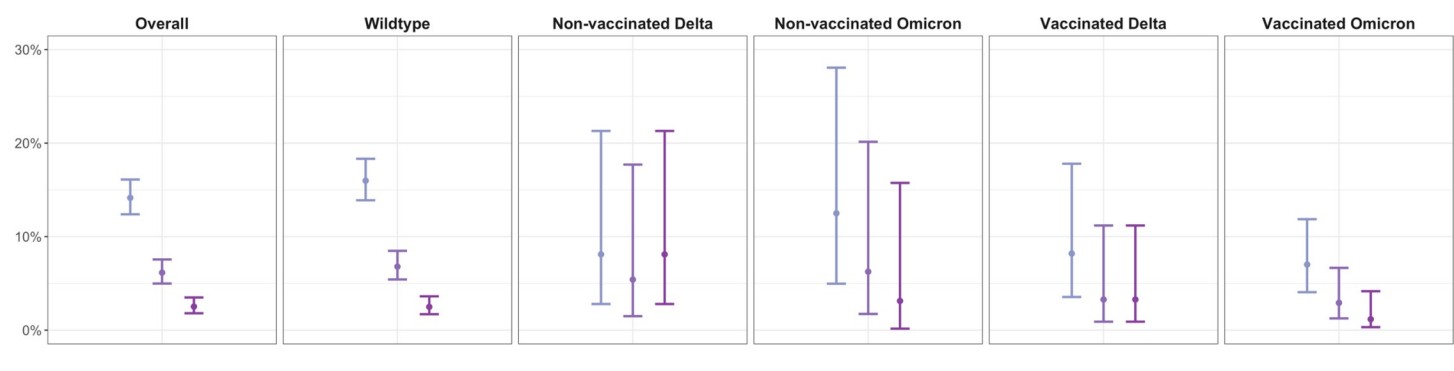

**Fig 2. Prevalence of post COVID-19 condition six months after infection at different levels of severity in terms of symptom count.** Analyses were stratified by SARS-CoV-2 variant and vaccination status. Points represent point estimate and error bars represent 95% Wilson confidence intervals for estimated proportions.

**Table 2. Results from multinomial logistic regression analyses of the association of SARS-CoV-2 variant and vaccination with severity of post COVID-19 condition in terms of symptom count.**

| Characteristic | 1–2 symptoms | | 3–5 symptoms | | ≥6 symptoms | |
|---|---|---|---|---|---|---|
| | OR (95% CI) | p-value | OR (95% CI) | p-value | OR (95% CI) | p-value |
| **Non-vaccinated Wildtype** | Ref. | | Ref. | | Ref. | |
| **Non-vaccinated Delta** | 0.50 (0.15–1.74) | 0.28 | 0.76 (0.15–3.72) | 0.73 | 3.91 (0.97–15.7) | 0.055 |
| **Non-vaccinated Omicron** | 0.80 (0.26–2.45) | 0.69 | 0.76 (0.14–4.08) | 0.75 | 1.95 (0.23–16.6) | 0.54 |
| **Vaccinated Delta** | 0.46 (0.18–1.18) | 0.11 | 0.49 (0.11–2.07) | 0.33 | 1.36 (0.31–6.02) | 0.69 |
| **Vaccinated Omicron** | 0.39 (0.21–0.74) | 0.0038 | 0.44 (0.16–1.19) | 0.11 | 0.50 (0.11–2.21) | 0.36 |

Legend: CI = confidence interval, OR = odds ratio, Ref. = reference group.

## Symptom clusters

Cluster analyses resulted in the identification of four clusters with different patterns of PCC-related symptoms (Fig 3). Based on predominant symptom patterns in each cluster, we categorised them into groups with diverse systemic (n = 219 of participants with PCC), neurocognitive (n = 47), cardiorespiratory (n = 23), and musculoskeletal symptoms (n = 19). Certain symptoms were prevalent in all clusters, such as fatigue, post-exertional malaise, headache, and taste or smell disturbances. In sensitivity analyses assuming five or six clusters, additional groups predominantly affected by gastrointestinal disturbances or hair loss, and by vertigo or dizziness were identified (S5 and S6 Figs). Across infections with different SARS-CoV-2 variants, we observed no clear differences in the proportion of participants belonging to each cluster (Fig 3). Female participants were more prevalent in the neurocognitive and cardiorespiratory clusters and older participants and individuals with comorbidities were more prevalent in the cardiorespiratory and musculoskeletal clusters, while the majority of participants with diverse systemic symptoms reported having one PCC-related symptom (S8 Table).

## Discussion

### Main findings in context

In this pooled analysis of 1350 SARS-CoV-2-infected individuals from two population-based cohorts, we found that infection with the Omicron variant and prior vaccination were associated with lower odds of PCC six months after infection compared to non-vaccinated, Wildtype-infected individuals. Meanwhile, the odds among non-vaccinated individuals infected with the Delta or Omicron variant were similar to those infected with Wildtype SARS-CoV-2. We found no differences in the reduction in odds of developing PCC between individuals having received 1–2 vaccine doses and those having received three vaccine doses, and between individuals last vaccinated more or less than six months prior infection. Compared to non-vaccinated individuals infected with Wildtype SARS-CoV-2, the severity of PCC symptoms was lower among vaccinated individuals infected with the Omicron variant, while a risk for developing PCC even of high severity was still present.

To our knowledge, this study is the first to simultaneously evaluate the impact of prior vaccination and infection with the Omicron variant on developing PCC up to six months after infection. In our study, the odds among non-vaccinated individuals infected with the Omicron variant were reduced by 13% compared to those infected by Wildtype SARS-CoV-2, corresponding to an absolute risk reduction of about 3 in 100 infected individuals. Among vaccinated individuals, the odds were reduced by 45% and 58% for Delta and Omicron infection, respectively, compared to non-vaccinated, Wildtype-infected individuals. This corresponds to an

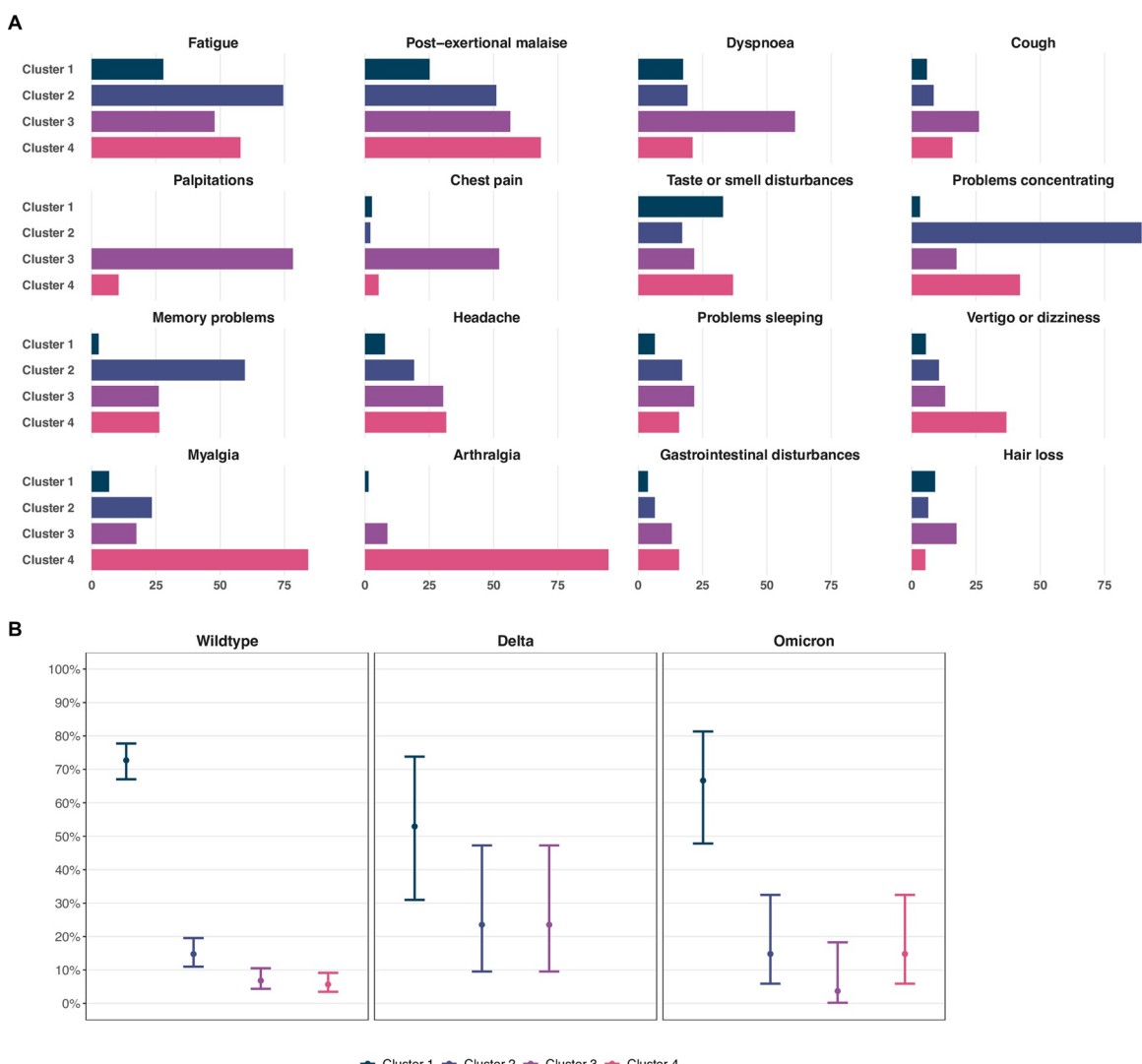

**Fig 3. Prevalence of specific post COVID-19 condition-related symptoms six months after SARS-CoV-2 infection across symptom clusters.** Four clusters of individuals with post COVID-19 condition at six months after infection were identified based on multiple correspondence and hierarchical cluster analyses, consisting of individuals with (1) diverse systemic symptoms and lower symptom count, and with (2) predominantly neurocognitive, (3) cardiorespiratory, or (4) musculoskeletal symptoms. Panel A depicts distributions of specific post COVID-19 condition-related symptoms across clusters. Panel B shows the proportion of individuals belonging to each cluster across infections with Wildtype, Delta, and Omicron SARS-CoV-2. Points represent point estimate and error bars represent 95% Wilson confidence intervals for estimated proportions.

absolute risk reduction in Omicron-infected of about 4 in 100 compared to vaccinated, Delta-infected individuals and of 10 in 100 compared to non-vaccinated, Wildtype-infected individuals. These estimates are in line with a relative risk reduction of 33% between infections with Omicron and prior variants at three months reported in a previous study [18]. While one further study reported no association of pooled Omicron and Delta infection with PCC compared to Wildtype infection at four weeks [8], another reported a stronger reduction in the odds of PCC between Omicron and Delta infection among vaccinated individuals [17]. The authors also found a stronger reduction in odds among those vaccinated more than three months prior infection compared to those vaccinated less than three months before. In our study, we did not identify a difference depending on whether the last vaccination was received more or less than

six months prior to infection. One previous study found a lower PCC-related symptom count among non-vaccinated individuals infected with the Alpha and Delta variants [19], but no other study investigated potential effects on severity of PCC. Further evidence from representative population-based studies is thus necessary to better estimate the reduction in the risk and severity of PCC in the longer-term while accounting for vaccination. While it cannot be excluded that future variants may bear a higher risk for PCC, this population-based study provides important early evidence on its longer-term risk and severity with the Omicron variant.

There is substantial heterogeneity between existing studies evaluating the effects of vaccination on PCC [5]. Our findings together with those from others imply that vaccination may reduce the risk of PCC by up to 50% [6–8,11,12,14–16], with an estimated reduction in the odds of 52% among Omicron-infected, and 35% among Delta-infected in our study. Despite important differences in study populations, study designs, analytical methods, and definitions of PCC across existing studies, our estimates are in line with previous evidence, albeit of smaller magnitude compared with one other study simultaneously investigating differences between SARS-CoV-2 variants and prior vaccination among non-hospitalised healthcare workers [8]. The accumulating evidence on potential preventive effects of vaccination on PCC has important implications for vaccination strategies and may be used to inform and positively influence individual decisions regarding booster vaccinations. However, a rigorous evaluation of the evidence, taking into account the substantial heterogeneity in study designs and the time periods during which they took place, is warranted prior to issuing recommendations regarding the prevention of PCC through vaccination.

In cluster analyses, we identified four distinct clusters of PCC-related symptoms across different variants, which we categorised as diverse systemic, neurocognitive, cardiorespiratory, and musculoskeletal symptoms. One study so far has investigated symptom clusters in the context of different SARS-CoV-2 variants [33]. The authors reported three main emerging PCC symptom clusters (i.e., central neurological, cardiorespiratory, and systemic/inflammatory), while the total number of identified clusters and cluster profiles varied across Wildtype, Alpha, and Delta variants. In our study, we found no substantial variation in the prevalence of symptom clusters across infections with different variants, but relevant variation in participant characteristics across clusters. Further studies have investigated the clustering of PCC-related symptoms, leading to the identification of similar groups with neurocognitive, cardiorespiratory, musculoskeletal and pain-related, and systemic or diverse symptoms [34,35]. Our findings are thus in line with existing evidence and suggest that some symptoms may be common to all presentations of PCC, while there may be distinct phenotypes of PCC that may also have different pathophysiological explanations or require different clinical management or treatment.

## Strengths and limitations

This study has several strengths, including the representative population-based sample and the prospective design, alongside evaluating PCC six months after infection. Meanwhile, several limitations have to be considered when interpreting the results. First, we pooled data from two closely aligned cohorts. Due to the different sampling, recruitment, and timeframes of data collection, there may still be residual confounding relating to differences between the participants in the two cohorts (e.g., socio-economic factors or behavioural aspects) that we could not fully account for in adjusted analyses. Second, selection may have occurred if participants with long-term symptoms were more likely to be enrolled or complete the questionnaire. However, in the Zurich SARS-CoV-2 Cohort, participants were enrolled prior to the possible occurrence of PCC, and there was minimal loss to follow-up. Corona Immunitas was a highly public, governmentally supported seroprevalence study using a random population sample, from which

we included all individuals with diagnosed infection. While some asymptomatically infected individuals in this study may not have sought testing and thus were excluded, we consider the probability of relevant selection bias regarding our findings to be low in both cohorts. Third, we relied on self-reported measures and could not perform a clinical validation of the relation of symptoms with initial SARS-CoV-2 infection or alternative diagnoses. While information bias cannot be excluded, we consider its potential effect on our findings to be minimal. In addition, it may be that some individuals in the Corona Immunitas Cohort had an asymptomatic or undiagnosed SARS-CoV-2 infection prior to the infection event evaluated in this analysis. Hence, the percentage of individuals with prior infection may be underestimated. Due to the design of the study, it was not possible to reliably ascertain prior infections using serological testing. Based on current evidence, it is unclear whether, to what extent, and in what direction any misclassification in our study could have biased our findings. Fourth, we did not have direct genetic data from viral sequencing and may have misclassified some participants infected with the Delta or Omicron variant based on our timeframe cut-offs. Fifth, we determined severity of PCC based on symptom count and EQ-VAS, of which the first may not be a direct correlate of severity and the latter may be influenced by baseline health status. While we tried to account for this in sensitivity analyses leading to broadly similar findings, results may still be confounded. Sixth, vaccinated participants were almost exclusively vaccinated with mRNA-based vaccines, so that our findings may not be fully generalisable to other vaccine types. Seventh, the study was not adequately powered to investigate differences between non-vaccinated individuals infected with Delta or Omicron and those infected with Wildtype SARS-CoV-2, and between individuals vaccinated once or twice and individuals that received three vaccine doses, leading to substantial statistical uncertainty. While estimates of greater precision from other studies are desirable, we consider the reliability of our findings as high due to the representative study populations. Last, the exploratory cluster analyses bear various limitations inherent in the structure and parametrisation of the model. Identified clusters are probabilistic and sample-bound, and may not coincide with phenotypes of PCC encountered in clinical practice. While our findings align with those by others, further research is necessary to more clearly identify different phenotypes based on their clinical and pathophysiological presentation.

## Conclusions

This study demonstrated that infection with the Omicron variant among individuals with prior vaccination is associated with a substantially reduced risk of PCC at six months post-infection compared to Wildtype SARS-CoV-2 infection among non-vaccinated individuals. While the risk of developing PCC appears to persist in the context of vaccination and novel variants, the risk reduction through vaccination was of greater magnitude than with infection by the Delta or Omicron variant. As the pandemic continues to evolve, vaccination will remain key in reducing the acute and long-term burden of SARS-CoV-2. Findings from this study underscore the significance of infection prevention and have important implications for public health messaging. Specifically, this information should be considered in vaccination campaigns and communication strategies, as well as for further vaccine development and the planning of public health measures for future pandemic waves.

## Supporting information

**S1 Fig. Flowchart of the enrolment, data collection, and inclusion of participants from the Zurich SARS-CoV-2 Cohort and from Phase 5 of the Corona Immunitas seroprevalence study.**
(DOCX)

**S2 Fig. Specific symptoms related to post COVID-19 condition across individuals infected with Wildtype, Delta, and Omicron SARS-CoV-2.**
(DOCX)

**S3 Fig. Results from sensitivity analysis of the association of Delta and Omicron SARS-CoV-2 infection, prior vaccination, and prior infection with post COVID-19 syndrome six months after SARS-CoV-2 infection, and of the association of vaccination with post COVID-19 syndrome stratified by number of received vaccine doses and timing of vaccination.**
(DOCX)

**S4 Fig. Results from sensitivity analyses regarding the prevalence of post COVID-19 condition six months after infection at different levels of severity.**
(DOCX)

**S5 Fig. Prevalence of specific post COVID-19 condition-related symptoms six months after SARS-CoV-2 infection across symptom clusters, based on a sensitivity analysis assuming five clusters.**
(DOCX)

**S6 Fig. Prevalence of specific post COVID-19 condition-related symptoms six months after SARS-CoV-2 infection across symptom clusters, based on a sensitivity analysis assuming six clusters.**
(DOCX)

**S1 Table. Eligibility criteria, recruitment timeframes, and assessments of the Zurich SARS-CoV-2 Cohort and Phase 5 of the Corona Immunitas seroprevalence study.**
(DOCX)

**S2 Table. List of the 23 post COVID-19 condition-related symptoms elicited in the Zurich SARS-CoV-2 Cohort and the Corona Immunitas seroprevalence study questionnaires.**
(DOCX)

**S3 Table. Detailed participant characteristics of Wildtype, Delta, or Omicron SARS-CoV-2-infected individuals from the Zurich SARS-CoV-2 Cohort and from Phase 5 of the Corona Immunitas seroprevalence study, stratified by study site.**
(DOCX)

**S4 Table. Comparison of participant characteristics of Phase 5 Corona Immunitas seroprevalence study participants in Zurich and Ticino, Switzerland, with non-included individuals (stratified by infection status).**
(DOCX)

**S5 Table. Results from sensitivity analyses of the association of SARS-CoV-2 variant and vaccination with severity of post COVID-19 condition based on multinomial logistic regression models, using symptom count restricted to six symptoms previously found to be in excess among those with post COVID-19 condition compared to the general population as severity categories.**
(DOCX)

**S6 Table. Results from sensitivity analyses of the association of SARS-CoV-2 variant and vaccination with severity of post COVID-19 condition based on multinomial logistic regression models, using current health status based on EQ-VAS scores as severity categories.**
(DOCX)

**S7 Table. Results from sensitivity analyses of the association of SARS-CoV-2 variant and vaccination with severity of post COVID-19 condition based on multinomial logistic regression, using current health status based on EQ-VAS scores as severity categories and restricting the analysis to individuals with no reported comorbidities at baseline to account for potential confounding by impaired baseline health status.**
(DOCX)

**S8 Table. Participant characteristics among infected individuals with post COVID-19 condition in the four clusters identified through cluster analysis.**
(DOCX)

**S1 File. Minimal dataset underlying the analyses in the study.**
(XLSX)

## Acknowledgments

The authors thank the study administration teams in Zurich and Ticino for their dedicated support of the study. Furthermore, the authors thank Hélène E. Aschmann and Anja Domenghino for their contribution to recruitment in the Zurich SARS-CoV-2 Cohort, and Sarah R. Haile for her statistical advice. Last, the authors thank the study participants for their valuable contribution to this project.

## Author Contributions

**Conceptualization:** Tala Ballouz, Dominik Menges, Marco Kaufmann, Rebecca Amati, Anja Frei, Viktor von Wyl, Jan S. Fehr, Emiliano Albanese, Milo A. Puhan.

**Data curation:** Tala Ballouz, Dominik Menges, Marco Kaufmann.

**Formal analysis:** Tala Ballouz, Dominik Menges.

**Funding acquisition:** Jan S. Fehr, Emiliano Albanese, Milo A. Puhan.

**Investigation:** Tala Ballouz, Dominik Menges, Marco Kaufmann, Rebecca Amati, Anja Frei, Viktor von Wyl, Jan S. Fehr, Emiliano Albanese, Milo A. Puhan.

**Methodology:** Tala Ballouz, Dominik Menges, Milo A. Puhan.

**Project administration:** Tala Ballouz, Dominik Menges, Marco Kaufmann, Rebecca Amati, Anja Frei, Viktor von Wyl.

**Supervision:** Jan S. Fehr, Emiliano Albanese, Milo A. Puhan.

**Validation:** Tala Ballouz, Dominik Menges, Marco Kaufmann.

**Visualization:** Tala Ballouz, Dominik Menges.

**Writing – original draft:** Tala Ballouz, Dominik Menges.

**Writing – review & editing:** Tala Ballouz, Dominik Menges, Marco Kaufmann, Rebecca Amati, Anja Frei, Viktor von Wyl, Jan S. Fehr, Emiliano Albanese, Milo A. Puhan.

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
