## [Decision Letter · Decision Letter 0]

21 Nov 2022

PONE-D-22-29701Post COVID-19 condition after Wildtype, Delta, and Omicron SARS-CoV-2 infection and prior vaccination: pooled analysis of two population-based cohortsPLOS ONE

Dear Dr. Puhan,

Thank you for submitting your manuscript to PLOS ONE. After careful consideration, we feel that it has merit but does not fully meet PLOS ONE’s publication criteria as it currently stands. Therefore, we invite you to submit a revised version of the manuscript that addresses the points raised during the review process.

We look forward to receiving your revised manuscript.

Kind regards,

Dong Keon Yon, MD, FACAAI

Academic Editor

PLOS ONE

Journal Requirements:

2. In ethics statement in the manuscript and in the online submission form, please provide additional information about the patient records/samples used in your retrospective study. Specifically, please ensure that you have discussed whether all data/samples were fully anonymized before you accessed them and/or whether the IRB or ethics committee waived the requirement for informed consent. If patients provided informed written consent to have data/samples from their medical records used in research, please include this information.

Additional Editor Comments:

Thank you for submitting your manuscript. The reviewers and I believe it is of potential value for our readers. However, the reviewers have raised a number of very important issues, and their excellent comments will need to be adequately addressed in a revision before the acceptability of your manuscript for publication in the Journal can be determined. We cannot guarantee that your revised paper will be chosen for publication; this would be solely based on how satisfactorily you have addressed the reviewer comments.

#1. Ref (4) is not a peer-reviewed article. Please cite a peer-reviewed article such as DOI: https://doi.org/10.54724/lc.2022.e10

Reviewers' comments:

Reviewer's Responses to Questions

**Comments to the Author**

1. Is the manuscript technically sound, and do the data support the conclusions?

Reviewer #1: Yes

Reviewer #2: Yes

Reviewer #3: Yes

Reviewer #4: Yes

2. Has the statistical analysis been performed appropriately and rigorously? 

Reviewer #1: Yes

Reviewer #2: I Don't Know

Reviewer #3: Yes

Reviewer #4: Yes

3. Have the authors made all data underlying the findings in their manuscript fully available?

Reviewer #1: Yes

Reviewer #2: Yes

Reviewer #3: Yes

Reviewer #4: Yes

4. Is the manuscript presented in an intelligible fashion and written in standard English?

Reviewer #1: Yes

Reviewer #2: Yes

Reviewer #3: Yes

Reviewer #4: Yes

5. Review Comments to the Author

Reviewer #1: Study is fine author has nicely compared two cohort. Though the findings from different studies are different, furthermore study is required for more generalizable of findings. It would have been more precise if the number in both cohort would be same or close enough.

Reviewer #2: The research reported in this manuscript is an important study that aims to add on to the building evidence surrounding the benefits of vaccination for COVID-19, and possibly for immune medicine in general.

The manuscript is well-written, easy to read and tables are informative. The study objectives are clearly set forth in the introduction and answered in the results lay out and discussion.

An important point of concern:

The description of the cohorts does not indicate to what extent seropravelent studies were employed to rule out possible prior asymptomatic infection amongst the Corona Immunitas cohort. Even though a percentage of that group is reported to have had previous infections, this appears to be based on self-report and not through an objective seroprevalence testing. To what extent was it possible that there was a higher percentage of previously asymptomatic infected persons ( and hence objectively unaware) in the Corona Immunitas group? Is it possible that previous (though asymptomatic) infection itself could confer some level of protection against PCC in itself, aside infection with Delta and Omicron variants? Does this not present a more serious confounding to the findings in the Corona Immunitas cohort?

Even though this is lightly discussed in limitations section, this could represent a more serious confounding and may require a more thorough discussion of any efforts to mitigate such or more clarity on the inability to objectively assess previous infection amongst Corona Immunitas cohort should be included when describing the cohort selection procedure. This may well assist readers to form a more balanced view of the study findings.

Reviewer #3: Dear authors,

I have read and reviewed the manuscript.

Please kindly find my recommendations below:

The manuscript has emphasized the importance of the preventive measures (especially vaccination). You have also mentioned the limitations of the study in an acceptable manner.

I would suggest the authors to discuss on the confounders and other variables which could not have been included in the study.

Recommendations in the last part might also be enlarged.

Regards,

Reviewer #4: Thank you for the opportunity to review this interesting manuscript by Ballouz and colleagues. I have reviewed the research work with great interest and enthusiasm. It is timely, and I strongly believe the findings will hugely contribute for the field. The manuscript is well-written, and the statistical methods are clearly described. I do have a few suggestions for the authors. The authors calculated odds ratio and reported risk of post COVID conditions as if they calculated risk ratio. Based on the their results, they are supposed to report odds of post COVID conditions not risk post COVID conditions. Their reporting need to be consistent with their analysis and results. The other thing, the supplementary table S3 shows that the Zurich SARS-CoV-2 Cohort had no history of Delta or Omicron variant infection, and had not been vaccinated. The authors may need to discuss the implication of this on their analysis and conclusion.

Overall, the manuscript is well written and I wish good luck for the authors.

Best regards,

Fekede

6. PLOS authors have the option to publish the peer review history of their article (what does this mean?). If published, this will include your full peer review and any attached files.

Reviewer #1: No

Reviewer #2: No

Reviewer #3: No

Reviewer #4: **Yes: **Fekede Asefa

---

## [Author Response · Author response to Decision Letter 0]

7 Dec 2022

Editor Comments:

Thank you for submitting your manuscript. The reviewers and I believe it is of potential value for our readers. However, the reviewers have raised a number of very important issues, and their excellent comments will need to be adequately addressed in a revision before the acceptability of your manuscript for publication in the Journal can be determined. We cannot guarantee that your revised paper will be chosen for publication; this would be solely based on how satisfactorily you have addressed the reviewer comments.

Thank you very much for your positive evaluation and your valuable time for assessing our manuscript. We address your and the reviewers' comments in the following paragraphs.

#1. Ref (4) is not a peer-reviewed article. Please cite a peer-reviewed article such as DOI: https://doi.org/10.54724/lc.2022.e10

While we agree that a peer-reviewed article would be ideal, there are none on the current number of SARS-CoV-2 cases. Furthermore, it has been common practice during the pandemic for many articles to refer to the Worldometer, John Hopkins University, or WHO COVID-19 dashboards as they provide regularly updated case numbers (for examples see: https://doi.org/10.1038/s41586-021-03914-4, https://doi.org/10.1016/S0140-6736(21)00947-8, https://doi.org/10.1016/S0140-6736(20)31142-9, http://doi.org/10.1056/NEJMoa2034577). The ICMJE reference samples also lists the possibility of citing electronic material: https://www.nlm.nih.gov/bsd/uniform_requirements.html. For this reason, we would prefer to keep the reference as it currently is.

The manuscript and Supplementary Information have been edited to meet the style requirements of PLOS ONE. Please find the changes highlighted within the text. As part of this revision, the Supplementary Information is now provided in individual files.

2. In ethics statement in the manuscript and in the online submission form, please provide additional information about the patient records/samples used in your retrospective study. Specifically, please ensure that you have discussed whether all data/samples were fully anonymized before you accessed them and/or whether the IRB or ethics committee waived the requirement for informed consent. If patients provided informed written consent to have data/samples from their medical records used in research, please include this information.

Our study is based on two prospective longitudinal cohorts in which all participants provided electronic (Zurich SARS-CoV-2 Cohort) or written (Corona Immunitas) informed consent prior to participation. This information is reported in the Methods section (Lines 85-100). We did not access patient records or use any samples collected in routine clinical practice.

Apologies for this discrepancy. We have now aligned the Funding Information and Financial Disclosure sections.

We have included a de-identified individual participant data as part of this submission (S15 File). All presented main analyses can be reproduced with the data provided.

We have removed the ethics statement from the Disclosures section and now only report it within the Methods section.

Captions for Supporting Information have been added at the end of the manuscript.

Reviewers' comments:

1. Is the manuscript technically sound, and do the data support the conclusions? The manuscript must describe a technically sound piece of scientific research with data that supports the conclusions. Experiments must have been conducted rigorously, with appropriate controls, replication, and sample sizes. The conclusions must be drawn appropriately based on the data presented. 

Reviewer #1: Yes

Reviewer #2: Yes

Reviewer #3: Yes

Reviewer #4: Yes

We thank the reviewers very much for their time and effort in evaluating our manuscript. There is no point to address with respect to this item.

2. Has the statistical analysis been performed appropriately and rigorously? 

Reviewer #1: Yes

Reviewer #2: I Don't Know

Reviewer #3: Yes

Reviewer #4: Yes

We can confirm that we have applied all due diligence with respect to the statistical analysis within the manuscript. Since the minimal dataset will be made available, it will be possible to verify all results.

3. Have the authors made all data underlying the findings in their manuscript fully available? The PLOS Data policy requires authors to make all data underlying the findings described in their manuscript fully available without restriction, with rare exception (please refer to the Data Availability Statement in the manuscript PDF file). The data should be provided as part of the manuscript or its supporting information, or deposited to a public repository. For example, in addition to summary statistics, the data points behind means, medians and variance measures should be available. If there are restrictions on publicly sharing data—e.g. participant privacy or use of data from a third party—those must be specified.

Reviewer #1: Yes

Reviewer #2: Yes

Reviewer #3: Yes

Reviewer #4: Yes

A minimal dataset allowing the reproduction of all results will be uploaded with the revised manuscript.

4. Is the manuscript presented in an intelligible fashion and written in standard English? PLOS ONE does not copyedit accepted manuscripts, so the language in submitted articles must be clear, correct, and unambiguous. Any typographical or grammatical errors should be corrected at revision, so please note any specific errors here.

Reviewer #1: Yes

Reviewer #2: Yes

Reviewer #3: Yes

Reviewer #4: Yes

Thank you. There is no point to address with respect to this item.

5. Review Comments to the Author. Please use the space provided to explain your answers to the questions above. You may also include additional comments for the author, including concerns about dual publication, research ethics, or publication ethics. (Please upload your review as an attachment if it exceeds 20,000 characters)

Reviewer #1:

Study is fine author has nicely compared two cohort. Though the findings from different studies are different, furthermore study is required for more generalizable of findings. It would have been more precise if the number in both cohort would be same or close enough.

Thank you very much for your comment. We agree that a larger sample size of individuals infected with Delta or Omicron would have been desirable and would have provided more precise estimates, particularly in the comparison of non-vaccinated individuals with Delta or Omicron compared to Wildtype SARS-CoV-2. However, in many populations worldwide the percentage of persons with infection but without vaccination has become so small that a high precision of estimates can only be achieved based on routine or surveillance data (which have their own limitations). Prospective cohort studies like ours would need to be very large to provide precise estimates. We have expanded on this point in the limitations section (lines 409-414):

“Seventh, the study was not adequately powered to investigate differences between non-vaccinated individuals infected with Delta or Omicron and those infected with Wildtype SARS-CoV-2, and between individuals vaccinated once or twice and individuals that received three vaccine doses, leading to substantial statistical uncertainty. While estimates of greater precision from other studies are desirable, we consider the reliability of our findings as high due to the representative study population and prospective follow-up.”

Reviewer #2:

The research reported in this manuscript is an important study that aims to add on to the building evidence surrounding the benefits of vaccination for COVID-19, and possibly for immune medicine in general.

The manuscript is well-written, easy to read and tables are informative. The study objectives are clearly set forth in the introduction and answered in the results lay out and discussion.

Thank you very much for your positive evaluation.

An important point of concern: The description of the cohorts does not indicate to what extent seropravelent studies were employed to rule out possible prior asymptomatic infection amongst the Corona Immunitas cohort. Even though a percentage of that group is reported to have had previous infections, this appears to be based on self-report and not through an objective seroprevalence testing. To what extent was it possible that there was a higher percentage of previously asymptomatic infected persons ( and hence objectively unaware) in the Corona Immunitas group? Is it possible that previous (though asymptomatic) infection itself could confer some level of protection against PCC in itself, aside infection with Delta and Omicron variants? Does this not present a more serious confounding to the findings in the Corona Immunitas cohort? 

Even though this is lightly discussed in limitations section, this could represent a more serious confounding and may require a more thorough discussion of any efforts to mitigate such or more clarity on the inability to objectively assess previous infection amongst Corona Immunitas cohort should be included when describing the cohort selection procedure. This may well assist readers to form a more balanced view of the study findings.

This is a very important point. We agree that it is possible that some individuals in the Corona Immunitas seroprevalence study may have had an asymptomatic or symptomatic but undiagnosed infection prior to the infection event used in the study. Unfortunately, serological testing could not be used to ascertain this since 1) around 80% were vaccinated (hence, we could not reliably distinguish prior infection and vaccination by anti-S antibodies, and anti-N antibodies are not detectable in all infections and wane relatively rapidly (https://doi.org/10.1038/s41467-022-32573-w)) and 2) we did not collect data or samples prior to the infection event. As you correctly state, it is thus not reliably possible to determine prior infection status in this study other than through the questionnaires.

One could hypothesize that prior infection confers some level of protection from PCC. Meanwhile, a prominent study recently has found the risk of SARS-CoV-2 infection-related complications to be increased (https://www.nature.com/articles/s41591-022-02051-3). For this reason, it is unclear whether, to what extent, and in what direction any misclassification in our study could have biased our findings. We agree that this is an important limitation and have added it more explicitly to the limitations section (lines 395-401):

"In addition, it may be that some individuals in the Corona Immunitas Cohort had an asymptomatic or undiagnosed SARS-CoV-2 infection prior to the infection event evaluated in this analysis. Hence, the percentage of individuals with prior infection may be underestimated. Due to the design of the study, it was not possible to reliably ascertain prior infections using serological testing. Based on current evidence, it is unclear whether, to what extent, and in what direction any misclassification in our study could have biased our findings."

Reviewer #3:

Dear authors,

I have read and reviewed the manuscript. Please kindly find my recommendations below:

The manuscript has emphasized the importance of the preventive measures (especially vaccination). You have also mentioned the limitations of the study in an acceptable manner.

I would suggest the authors to discuss on the confounders and other variables which could not have been included in the study.

Recommendations in the last part might also be enlarged. Regards,

Thank you very much for your valuable feedback. As we mention in the limitation section, it is possible that there may be residual confounding in our study relating to differences between the two populations. We have now slightly extended on this point to highlight it more (lines 380-383):

“Due to the different sampling, recruitment, and timeframes of data collection, there may still be residual confounding relating to differences between the participants in the two cohorts (e.g., socio-economic factors or behavioural aspects) that we could not fully account for in adjusted analyses.”

The conclusion has also been slightly expanded (lines 422-433):

“This study demonstrated that infection with the Omicron variant among individuals with prior vaccination is associated with a substantially reduced risk of PCC at six months post-infection compared to Wildtype SARS-CoV-2 infection among non-vaccinated individuals. While the risk of developing PCC appears to persist in the context of vaccination and novel variants, the risk reduction through vaccination was of greater magnitude than with infection by the Delta or Omicron variant. As the pandemic continues to evolve, vaccination will remain key in reducing the acute and long-term burden of SARS-CoV-2. Findings from this study underscore the significance of infection prevention and have important implications for public health messaging. Specifically, this information should be considered in vaccination campaigns and communication strategies, as well as for further vaccine development and the planning of public health measures for future pandemic waves.“

Reviewer #4:

Thank you for the opportunity to review this interesting manuscript by Ballouz and colleagues. I have reviewed the research work with great interest and enthusiasm. It is timely, and I strongly believe the findings will hugely contribute for the field. The manuscript is well-written, and the statistical methods are clearly described. I do have a few suggestions for the authors. The authors calculated odds ratio and reported risk of post COVID conditions as if they calculated risk ratio. Based on the their results, they are supposed to report odds of post COVID conditions not risk post COVID conditions. Their reporting need to be consistent with their analysis and results. The other thing, the supplementary table S3 shows that the Zurich SARS-CoV-2 Cohort had no history of Delta or Omicron variant infection, and had not been vaccinated. The authors may need to discuss the implication of this on their analysis and conclusion.

Overall, the manuscript is well written and I wish good luck for the authors. Best regards, Fekede

Thank you very much for your evaluation and helpful comments. You are right that since we calculated odds ratios, we should be report the results as odds rather than risks. We now consistently use odds in the Results section of the manuscript. However, we kept the interpretation of the results in some parts of the discussion using the term "risk" where it refers to the estimated absolute risk reduction, or more generally to the existing evidence for easier interpretation. Regarding your comment on the participants of the Zurich SARS-CoV-2 Cohort, the enrolment timeframe of the study covered a period where Wildtype was the predominant strain and vaccination was not yet available in Switzerland. While it certainly would have been preferable to have individuals with different vaccination status and infected with different strains within the same cohort for the analyses, this was not possible. However, since participants in both cohorts were drawn at random, we do not perceive this to have significant implications on our data, other than that there may be residual confounding relating to differences in the participants of the Zurich SARS-CoV-2 Cohort and Corona Immunitas which we did not adjust for (mentioned in lines 379-383):

“First, we pooled data from two closely aligned cohorts. Due to the different sampling, recruitment, and timeframes of data collection, there may still be residual confounding relating to differences between the participants in the two cohorts (e.g., socio-economic factors or behavioural aspects) that we could not fully account for in adjusted analyses.” 

6. PLOS authors have the option to publish the peer review history of their article (what does this mean?). If published, this will include your full peer review and any attached files. Do you want your identity to be public for this peer review? For information about this choice, including consent withdrawal, please see our Privacy Policy.

Reviewer #1: No

Reviewer #2: No

Reviewer #3: No

Reviewer #4: Yes: Fekede Asefa

There is no point to address with respect to this item.

---

## [Decision Letter · Decision Letter 1]

24 Jan 2023

Post COVID-19 condition after Wildtype, Delta, and Omicron SARS-CoV-2 infection and prior vaccination: pooled analysis of two population-based cohorts

PONE-D-22-29701R1

Dear Dr. Puhan,

We’re pleased to inform you that your manuscript has been judged scientifically suitable for publication and will be formally accepted for publication once it meets all outstanding technical requirements.

Kind regards,

Dong Keon Yon, MD, FACAAI

Academic Editor

PLOS ONE

Additional Editor Comments (optional):

This is an excellent paper.

Reviewers' comments:

Reviewer's Responses to Questions

**Comments to the Author**

1. If the authors have adequately addressed your comments raised in a previous round of review and you feel that this manuscript is now acceptable for publication, you may indicate that here to bypass the “Comments to the Author” section, enter your conflict of interest statement in the “Confidential to Editor” section, and submit your "Accept" recommendation.

Reviewer #2: All comments have been addressed

Reviewer #3: All comments have been addressed

Reviewer #4: All comments have been addressed

2. Is the manuscript technically sound, and do the data support the conclusions?

Reviewer #2: Yes

Reviewer #3: Yes

Reviewer #4: Yes

3. Has the statistical analysis been performed appropriately and rigorously? 

Reviewer #2: I Don't Know

Reviewer #3: Yes

Reviewer #4: Yes

4. Have the authors made all data underlying the findings in their manuscript fully available?

Reviewer #2: Yes

Reviewer #3: Yes

Reviewer #4: Yes

5. Is the manuscript presented in an intelligible fashion and written in standard English?

Reviewer #2: Yes

Reviewer #3: Yes

Reviewer #4: Yes

6. Review Comments to the Author

Reviewer #2: (No Response)

Reviewer #3: Dear authors,

My recommendations are covered in the revized manuscript.

This revision will hopefully help to understand the content better.

Regards,

Reviewer #4: Dear authors,

You have addressed all the the comments I provided to you.

I wish you all the best.

7. PLOS authors have the option to publish the peer review history of their article (what does this mean?). If published, this will include your full peer review and any attached files.

Reviewer #2: **Yes: **Joshua Appiah Arthur

Reviewer #3: No

Reviewer #4: **Yes: **Fekede Asefa

---

## [Editor Report · Acceptance letter]

26 Jan 2023

PONE-D-22-29701R1 

Post COVID-19 condition after Wildtype, Delta, and Omicron SARS-CoV-2 infection and prior vaccination: pooled analysis of two population-based cohorts 

Dear Dr. Puhan:

I'm pleased to inform you that your manuscript has been deemed suitable for publication in PLOS ONE. Congratulations! Your manuscript is now with our production department. 

Kind regards, 

on behalf of

Dr. Dong Keon Yon 

Academic Editor

PLOS ONE